# Relationship between Phase Angle and Physical Activity Intensity among Community-Dwelling Older Adults in Japan: A Cross-Sectional Study

**DOI:** 10.3390/healthcare12020167

**Published:** 2024-01-10

**Authors:** Daiki Nakashima, Keisuke Fujii, Yoshihito Tsubouchi, Yuta Kubo, Kyosuke Yorozuya, Kento Noritake, Naoki Tomiyama, Soma Tsujishita, Terufumi Iitsuka

**Affiliations:** 1Department of Rehabilitation, Faculty of Health Science, Naragakuen University, 3-15-1, Nakatomigaoka, Nara 631-8524, Nara, Japaniitsuka@naragakuen-u.jp (T.I.); 2Department of Rehabilitation Occupational Therapy Course, Faculty of Health Science, Suzuka University of Medical Science, 1001-1, Kishioka, Suzuka 510-0293, Mie, Japan; fujii@suzuka-u.ac.jp; 3Division of Occupational Therapy, Faculty of Rehabilitation and Care, Seijoh University, 2-172 Fukinodai, Tokai 476-8588, Aichi, Japan; kubo-yu@seijoh-u.ac.jp (Y.K.); yorozuya@seijoh-u.ac.jp (K.Y.); tomiyama@seijoh-u.ac.jp (N.T.); 4Department of Rehabilitation, Faculty of Health Sciences, Nihon Fukushi University, Higashihaemi, Handa 475-0012, Aichi, Japan; noritake-k@n-fukushi.ac.jp; 5Division of Physical Therapy, Faculty of Rehabilitation and Care, Kobe International University, 9-1-6 Kouyou, Higashinada, Kobe 658-0032, Hyogo, Japan; tsujishita@kobe-kiu.ac.jp

**Keywords:** exercise intensity, physical health, body composition, older people, geriatrics

## Abstract

This cross-sectional study aimed to determine the association between phase angle (PhA) and physical activity intensity in community-dwelling older Japanese adults. The intensity and time of physical activity for predicting high PhA were also examined. This study involved 67 community-dwelling older adults (mean age: 78.3 ± 5.5; female: 83.6%). We measured the physical activity and body composition of the participants. Physical activity was measured using a triaxial accelerometer, and light-intensity physical activity (LPA) and moderate-to-vigorous intensity physical activity (MVPA) hours per day were calculated from the results. Body composition was measured using Inbody S10, and the PhA was calculated from the measurements of the right side of the body. Bayesian statistical modeling revealed an association between PhA and MVPA (β = 0.256; *p* = 0.022; 95% Bayesian confidence interval [CI] = 0.001, 0.012), but not LPA (β = −0.113; *p* = 0.341; 95% Bayesian CI = −0.002, 0.001), even after adjustment for confounders. The cutoff value of MVPA predicting high PhA, calculated by the receiver operator characteristic curve, was 19.7 min/d (sensitivity = 0.906; specificity = 0.429). These results can be used to develop strategies to increase PhA in older adults and suggest that MVPA is important in this population.

## 1. Introduction

Bioelectrical impedance analysis (BIA) is widely used as a simple and non-invasive method for measuring body composition. Impedance, a generic term for the opposition of the body to alternating currents, comprises resistance (R) and reactance (Xc). The phase angle (PhA), which is directly calculated from BIA measurements, is considered to indicate cellular health [1] and has attracted attention as an easily measurable physical health indicator for older adults. A lower PhA has been suggested as a predictor of frailty and is associated with mortality in older adults [2]. Therefore, it is necessary to improve PhA from the perspective of health promotion among older adults.

Recently, the importance of physical activity in maintaining and improving PhA in older adults has been highlighted [3]. Physical activity is divided into exercise-related and non-exercise-related activities [4], with different intensities depending on the activity. It is necessary to identify the appropriate exercise intensity for older adults to effectively enhance their PhA, and hence, physical health. On the other hand, lower PhA has been reported in clinical populations compared to that in healthy populations [5,6]. Additionally, in a study of three racial or ethnic groups (non-Hispanic white, non-Hispanic black, and Mexican American), it was found that PhA differed by race and ethnicity [7]. Furthermore, since Asians tend to have lower PhA than Caucasians tend to [8,9], it can be expected that the results will vary depending on the characteristics of the population studied. In light of these reports, it is necessary to consider health status and race. However, studies investigating the association between PhA and physical activity intensity in healthy older adults in Japan are limited. A cross-sectional study of older Japanese adults (mean age 75.2 years; mean PhA = 5.2 ± 0.8°) reported that moderate-to-vigorous physical activity (MVPA) was associated with higher PhA [10].

Physical activity is affected by the residential area and living environment. Although the research setting of a previous study was a rural region [11], physical activity and lifestyle may differ depending on the residential area (rural or urban) and neighborhood environment [12,13,14]. In addition, a study of East Asians (age range 30–79 years) showed that body mass index (BMI) differed between urban and rural areas [15], and although the age of the participants was different from that of this study, regional differences in body composition have also been reported among Japanese adolescents [16]. Based on these factors, regional characteristics (rural or urban) are sometimes included as adjustment variables in surveys dealing with body composition [17]. Therefore, accumulating studies from multiple regions is important for understanding the contribution of physical activity to PhA. Furthermore, it has been reported that age, sex, BMI, nutrition status, muscle strength (grip strength), walking speed, balance ability, skeletal muscle mass index (SMI), and muscle quality also affect PhA [1,18,19,20,21]. However, to the best of our knowledge, analyses that consider these confounders are insufficient, and the evidence for the effectiveness of physical activity would be stronger if the association between PhA and physical activity were found even after adjusting for these variables.

Additionally, since PhA is also associated with mortality in older adults, increasing PhA is an important effort to prevent disability and reduce the risk of mortality. The risk of incident disability related to PhA has been associated with physical activity, with higher levels of MVPA and light-intensity physical activity (LPA) in bouts of <10 min in older adult Japanese subjects being associated with a lower risk of functional impairment [22,23]. From this, it is inferred that physical activity intensity and times may also be related to PhA, and an association between MVPA and PhA has been reported in certain community-dwelling older adults [9]. However, it is not clear what kind of MVPA predicts high PhA. Identifying its cutoff value will provide useful information for improving the health of older adults.

Therefore, this study aimed to determine the association between PhA and the intensity of physical activity in community-dwelling older Japanese adults. In addition, we examined the intensity and time of physical activity required to predict high PhA. Based on trends in previous studies, we hypothesized that a higher PhA would be associated with moderate- or high-intensity physical activity.

## 2. Materials and Methods

### 2.1. Materials

#### 2.1.1. Sample Size Calculation

Regarding the number of explanatory variables in the multiple regression analysis, ten times the number of explanatory variables has been suggested as a guide [24]. In this study, age, sex, BMI, SMI, nutrition status, grip strength, balance ability, gait speed, and muscle quality were used as adjustment variables [9,18,19,20,21]. If LPA, MVPA, and adjustment variables are included, the total number of independent variables is assumed to be ten. Therefore, the target sample size was set to 110.

#### 2.1.2. Recruitment of Research Participants and Selection of Participants for Analysis

This cross-sectional study included 99 participants of health screening programs conducted from November 2022 to September 2023. The study area was the city of Nara, Nara Prefecture, Japan. Nara is located in the Kansai region of Japan and is the capital of the Nara Prefecture. Nara has a population of 349,948 (1 November 2023) [25], a population density of 1275 (2021) [26], and an aging rate of 31.9% (2023) [27]. The exclusion criteria for this study were the inability to give consent to participate in the study, being younger than 65 years of age, use of a pacemaker, a history of cerebrovascular disease or orthopedic surgery, a certification for requiring support or nursing care, and missing measurement items. The number of participants analyzed in this study is shown in Figure 1. Of the 99 participants, 27 met the exclusion criteria for this study or could not give consent to wear the accelerometer. Therefore, 72 participants were given accelerometers, but five participants were unable to wear the accelerometer for more than 10 h per day and more than 3 days per week, leaving 67 participants for the final analysis of this study. Participant recruitment was requested by the Regional Comprehensive Support Center in Nara, and materials for the health checkup program were distributed. This health checkup program was not a regular checkup activity, and we explained the content of our research to local comprehensive support centers in Nara City and recruited participants from communities that agreed to our research activities. This study was conducted in accordance with the Strengthening the Reporting of Observational Studies in Epidemiology Statement.

### 2.2. Ethical Considerations

In this study, all participants were given a detailed description of the study verbally and in writing and were asked to sign a consent form. This study was approved by the Ethics Review Committee of the authors’ affiliated organization (approval number: 4-H026).

### 2.3. Measurement Variables

#### 2.3.1. Participant Characteristics

Age, sex, household composition (Do you live alone? [Yes or No]), years of education, and medical and surgical histories were ascertained using questionnaires. Height (PA-200, UCHIDA, Tokyo, Japan) and weight (DP-7900PW, Yamato, Hyogo, Japan) were measured for all participants. BMI was calculated from the values obtained from body measurements using the following formula: BMI (kg/m^2^) = weight (kg)/height (m)^2^.

#### 2.3.2. Physical Activity

Physical activity was measured using an Active Style Pro (HJA-750C, Omron, Kyoto, Japan). The Active-style Pro has the same level of measurement accuracy as energy expenditure measured by the double-labeled water method [28] and is suitable for measuring physical activity on free-living days. Accelerometers were instructed to be worn for a minimum of 10 h per day for 1 week, except for bathing and other activities where there was a risk of submersion. An accelerometer was attached to the front of the left lumbar region. We distributed a leaflet on equipment precautions and provided individual verbal and practical demonstration explanations to each participant regarding the method of wearing the accelerometers. Furthermore, the accelerometer was set in such a way that only the clock could be viewed, and the participants could not check their step counts. The accelerometers were collected by mail and the data was captured on a personal computer using an Omron USB communication tray (HHX-IT4, Omron, Kyoto, Japan). A macro program (ver. 190829), developed and distributed by the Japan Physical Activity Research Platform, was used for accelerometer data processing [29]. LPA and MVPA times were extracted from the data processed using the macro program, and the mean values per day were calculated and used in the analysis. In this study, LPA was defined as 1.6 Mets to 2.9 Mets of activity, and MVPA was defined as 3 Mets or more of activity [30]. In addition, only those who wore the device for at least 10 h per day for at least 3 days were included in the analysis [31].

#### 2.3.3. Body Composition

Body composition was measured using an Inbody S10 (Inbody Japan, Tokyo, Japan) before all physical function measurements were performed. Furthermore, sufficient rest time was ensured before body composition measurements. All metal objects attached to the participants were removed before the measurements began. The limb to be measured was positioned supine, and electrodes were attached to the thumb, middle finger, and ankle joints bilaterally [32]. The PhA was calculated from the measurements of the right side of the body using the following formula: PhA (°) = (Xc × R) × 180°/π. The higher PhA shows better cellular function [1]. In this study, we defined low PhA as ≤4.95° in males and ≤4.35° in females, using the cutoff value of increased risk of incident disability occurrence in older adults as a reference [19,21].

The SMI was calculated by dividing the skeletal muscle mass of the extremities by the square of the height (SMI [kg/m^2^] = appendicular skeletal muscle mass [kg]/height [m]^2^) [32,33,34].

#### 2.3.4. Global Leadership Initiative on Malnutrition (GLIM) Criteria

The GLIM criteria are international standards for the diagnosis of malnutrition. Three phenotypic criteria (weight loss, low BMI, and reduced muscle mass) and two etiologic criteria (reduced food intake or assimilation and inflammation or disease burden) are assessed, and participants are considered malnourished if at least one phenotypic criterion and one etiologic criterion are met [35].

#### 2.3.5. Grip Strength

Grip strength was measured using a Smedley-type digital grip-strength meter (GROP-D; TAKEI, Niigata, Japan). The measurement position included the participant standing, with the elbow joint in extension and the arm lowered to the side of the body for two measurements. The highest values of the two measurements were used in this study.

#### 2.3.6. Short Physical Performance Battery (SPPB)

The SPPB is a physical performance test consisting of balance, gait speed, and chair-stand test [36,37]. Balance tests were performed in a side-by-side stand, semi-tandem stand, and tandem stand for 10 s each. Gait speed was recorded as the time taken to walk 4 m at normal gait speed. The chair-stand test involved the participant starting in a seated position on a chair, and the time for five repetitions of standing up and sitting down was measured. Each item was scored from 0 to 4, with the total score ranging from 0 to 12 points [37]. In this study, the gait speed per meter was calculated from the 4 m gait speed [18].

#### 2.3.7. Muscle Quality

Echo intensity, which reflects intramuscular adipose tissue and is frequently used to assess muscle quality, was used to evaluate muscle quality [38]. Echo intensity was calculated using ultrasound images taken in B mode (gain 85 dB, frequency 8.4 MHz, depth 4.0–7.0 cm) of an ultrasound system (Toshiba, Viamo SSA-640, Tokyo, Japan). The measurement muscle was the rectus femoris of the dominant leg, and the measurement limb was positioned supine on a bed with the ankle joint in the neutral position. Two evaluators, trained in echo intensity measurements, were responsible for the measurements to avoid measurement errors. The linear array probe was gently applied perpendicularly to the skin using a water-soluble permeable gel to prevent shape changes in the muscle due to probe pressure. The echo intensity was calculated from the ultrasound image results using the ImageJ software (version 1.53, National Institutes of Health, Bethesda, MD, USA). Echo intensities were obtained using 8-bit grayscale analysis, with the mean echo intensity of the manually enclosed area analyzed and expressed as a value from 0 to 255 (black = 0; white = 255) [38]. Lower echo intensity values indicated a better condition. In this study, the mean value of the echo intensity calculated from three ultrasound images was used for analysis.

### 2.4. Statistical Analysis

Means and standard deviations were calculated for continuous variables, and proportions were calculated for categorical variables. Pearson’s correlation coefficient was used to confirm the association between PhA and physical activity, age, BMI, SMI, grip strength, balance ability (SPPB balance tests), gait speed, and muscle quality. Spearman’s rank correlation coefficient was used to confirm the association between PhA and sex and nutrition status (GLIM criteria).

Ten independent variables, including eight confounding factors, were included in this study. Therefore, considering the possibility that the target sample size might not have been reached, we decided that Bayesian statistics, which can provide a stable estimation even with small sample sizes, should be used and adopted as the analytical method [39,40]. Bayesian statistics use a prior distribution and combine Bayesian estimation with the Markov chain Monte Carlo method, which is a random number generation algorithm, to generate parameters estimated as a posterior distribution from the acquired data. Bayesian statistics were selected for the analysis as the study focused on older adults in a specific area (one city), allowing the generation of a posterior distribution that better reflected the characteristics of this area.

As the dependent variable PhA is a continuous variable, modeling was conducted assuming a normal distribution. The settings were as follows: chain 4, iteration 3000, warm-up 2000, and thin 1. The prior distribution was assumed to be uninformed. In the convergence determination, convergence to the steady state was considered to have occurred when the estimated posterior distribution Rhat was less than 1.05 [40].

Age, sex, BMI, SMI, nutrition status (GLIM criteria), grip strength, balance ability (SPPB balance test), gait speed, and muscle quality were selected as adjustment variables. The models assumed a bivariate analysis model with PhA as the dependent variable and LPA and MVPA as independent variables (Model 1) and a multivariate analysis model that included confounding factors in Model 1 (Model 2). The widely applicable information criterion (WAIC) of both models was compared and the one with a lower value was adopted as the better model (the model more predictive of the phenomenon) [41]. WAIC is a measure for predicting true distribution; the closer the number is to zero, the better the model.

The results of the Bayesian statistical analysis were confirmed using Expected A Posteriori (EAP) and 95% Bayesian confidence intervals (CI). The 95% Bayesian CI in Bayesian statistics has a meaning similar to the 95% CI in conventional statistics and is interpreted as significant if the value does not contain zero (allowing for an interpretation independent of the *p*-value). In this study, t-values, standardized regression coefficients (β), and *p*-values were also calculated as indices of judgment, in analogy with conventional statistics.

To determine the cutoff value of MVPA time for predicting high PhA in older adults, a receiver operator characteristic (ROC) curve was used to calculate the area under the curve (AUC). The cutoff value was defined as the point with the highest sensitivity and specificity.

SPSS Statistics version 25.0 (IBM, Armonk, NY, USA) was used for descriptive statistics, Pearson’s correlation coefficient analysis, Spearman’s rank correlation coefficient analysis, and ROC analysis. Statistical significance was set at *p* < 0.05. R software version 4.0.5 (R Foundation for Statistical Computing, Vienna, Austria) with the RStan (version 2.21.2), loo (version 2.4.1), and brms (version 2.15.0) packages was used for Bayesian statistical analysis.

## 3. Results

### 3.1. Participants’ Characteristics

The participants’ characteristics are shown in Table 1. The mean age of the 67 participants was 78.3 ± 5.5 years and 83.6% were women. The mean years of education was 13.1 ± 2.1 years, and 44.8% of the participants were older adults living alone. The mean daily time of accelerometer wearing was 850.9 ± 93.9 min/d, the mean LPA time was 346.3 ± 84.0 min/d, and the mean MVPA time was 38.0 ± 23.2 min/d. The mean PhA was 4.4 ± 0.6°.

### 3.2. Correlation of PhA with LPA and MVPA

Table 2 shows the correlation between PhA and each variable. Significant negative correlations were found between PhA and age (r = −0.343; *p* < 0.01), sex (r = −0.526; *p* < 0.01), and muscle quality (r = −0.25; *p* = 0.041), while SMI (r = 0.581; *p* < 0.01) and grip strength (r = 0.645; *p* < 0.01) showed a significant positive correlation. BMI also showed a significant correlation trend with PhA (r = 0.204; *p* = 0.099). On the other hand, nutrition status (r = −0.176; *p* = 0.155), balance ability (r = −0.011; *p* = 0.927), and gait speed (r = −0.068; *p* = 0.583) showed no significant association.

Figure 2 and Figure 3 show the correlation between PhA and physical activity according to intensity. PhA and LPA were not significantly correlated (r = −0.089; *p* = 0.475); however, PhA and MVPA were significantly correlated (r = 0.305; *p* = 0.012). 

### 3.3. Model Selection

Bayesian statistical modeling produced all models as predictive distributions that approximated the true distribution (Rhat < 1.05). Models 1 and 2 produced a WAIC of 115.001 and 94.017, respectively, with Model 2 being selected as the better model.

### 3.4. Relationship between PhA and Physical Activity (Bayesian Statistics)

Table 3 presents the results of the Bayesian statistics. In the bivariate analysis model with PhA as the dependent variable and LPA and MVPA as independent variables (Model 1), only MVPA was significantly associated; LPA was not. The multivariate analysis model (Model 2) confirmed the association between PhA and MVPA, even after adjusting for confounders (β = 0.256; *p* = 0.022; 95% Bayesian CI = 0.001, 0.012).

### 3.5. ROC Analysis

ROC analysis showed AUC = 0.653; *p* = 0.032; 95% CI = 0.520, 0.785. Cut-off MVPA times were 19.7 min/d (sensitivity = 0.906; specificity = 0.429) (Figure 4).

## 4. Discussion

In this study, we examined the relationship between PhA and physical activity intensity in Japanese older adults to obtain useful information for establishing strategies to increase PhA in older adults. Bayesian statistics results showed an association between PhA and physical activity among community-dwelling older adults in Japan and found a significant association between PhA and MVPA, even after adjusting for age, sex, BMI, SMI, nutrition status, grip strength, balance ability (SPPB balance tests), gait speed, and muscle quality. The cutoff value for physical activity to predict high PhA was calculated by ROC analysis and found that MVPA of at least 19.7 min per day may be required.

Physiological and pathological conditions (age, sex, BMI, and race) should be considered when measuring PhA [42]. We targeted older Japanese adults as there have been limited studies focusing on PhA and physical activity intensity in this population. Yamada et al. found an association between PhA and MVPA in Japanese adults (age range: 32–69 years) after adjusting for age, sex, height, body fat percentage, body cell mass, and leg muscle power [43]. In a study by Asano et al., in which the participants were limited to older adults, PhA was found to be associated with MVPA after adjusting for age, sex, BMI, years of education, self-rated economic status (very poor/a little poor/normal/little good/very good), alcohol consumption, smoking status, history of diabetes and cancer, and muscle mass [10]. In this study, muscle quality, which has recently been indicated to be related to PhA, was added to the adjusted variables in previous studies and analyzed using a Bayesian statistical model. The results revealed that MVPA was associated with PhA, which is similar to the findings of previous studies. Moreover, physical activity is affected by the residential area and living environment. The target area of this study was the capital of Nara Prefecture, which has urban functions. Therefore, it has different characteristics from rural areas, which was the object of the previous study. Furthermore, comparing the major industries in each region–1.3% vs. 5.7% (Nara vs. previous study area) in the primary industry and 16.7% vs. 25.0% (Nara vs. previous study area) in the secondary industry [44,45]—it is predicted that the living environments are also different. MVPA was related to PhA even in older adults who belonged to regions where such characteristics were different; hence, this study enhances the scientific evidence of the contribution of MVPA to PhA.

However, LPA was not associated with PhA in the participants in this study. Unlike conventional statistics, Bayesian statistics allow the null hypothesis to be “adopted” [46]. Because the previous study used conventional statistical methods, the null hypothesis for the association between LPA and PhA, which did not show a significant difference, could not be adopted and thus could not be clarified. Therefore, using Bayesian statistics, the present study clarified the possibility that LPA is not associated with PhA in older adults. The effect of training on the PhA has been confirmed mainly with exercises that do not belong to the LPA group, such as resistance training [47]. In addition, although the relationship between muscle mass and the PhA has been reported [21], it has also been reported that older adults who participate in recreational activities (golfing, gardening, tennis, and/or cycling) more than three times a week together with activities of daily living (walking, grocery shopping, and gardening) could reduce their decline in muscle mass with aging more than inactive older adults [48]. Uemura et al. investigated the relationship between PhA and physical activity level (PAL) and found that PhA and PAL were related regardless of sex [49]. Because PAL indicates the mean intensity of daily activities, older adults who live with high activity intensity can be interpreted as having a higher PhA. Although it was not possible to identify a causal relationship in this study, both its results and trends from previous studies suggest that an increase in MVPA is important for improving PhA. In the future, longitudinal studies should be conducted to examine the causal relationship between MVPA and PhA.

The correlation between PhA and each of the variables in this study showed that age, sex, and muscle quality had significant negative correlations, while SMI and grip strength had significant positive correlations. Furthermore, BMI also showed a significant correlation trend with PhA These relationships were similar to those found in previous studies [9,18,21]. On the other hand, there was no correlation between PhA and nutrition status, balance ability (SPPB balance tests), and gait speed. The GLIM criteria for nutrition status were used in this study, which indicated that only 10% of the participants were undernourished, suggesting that the results were influenced by the fact that most of the participants were in good nutritional condition. In addition, most of the participants had perfect scores on the balance test and gait speed evaluation, which are related to SPPB, and the low variability of the measured values is likely to have influenced the results. In addition, since the Timed Up and Go test was used as the balance test in the previous study [20], the specificity of the test may have influenced the results.

In this study, ROC analysis suggested that engaging in MVPA for more than 19.7 min per day may predict high PhA. This result indicates that approximately 140 min or more of MVPA per week is required. This value is similar to the amount of exercise (at least 150–300 min of moderate-intensity aerobic physical activity or at least 75–150 min of vigorous-intensity aerobic physical activity) recommended by the World Health Organization’s Guidelines on Physical Activity and Sedentary Behavior [50]. However, caution must be exercised in interpreting the results of this study. The AUC calculated from the ROC analysis was 0.653, leaving the question of accuracy. Since the number of participants in this study was small, it is necessary to increase the number of participants to improve the accuracy of the analysis and to examine the reference values for age and sex in the future.

In this study, we found an association between PhA and MVPA in older Japanese adults living in a medium-sized city after adjusting for important confounders. However, this study had several limitations. First, as this was a cross-sectional study, the causal relationship between PhA and MVPA could not be clarified. In the future, it is necessary to conduct longitudinal studies and observe the effects of MVPA on PhA over a longer period. In addition, given the low rate of compliance with physical activity guidelines among older adults [51], interventional studies on increasing MVPA among older adults are needed. Second, the sample size was small. This issue was addressed using Bayesian statistical modeling. However, Bayesian statistics reflect the attributes of the target population, and it is unclear whether the results of this study are applicable to other regions. It is necessary to collect data on different participants from other regions and perform a Bayesian update based on the results. To recruit enough number of participants for future studies, we will conduct outreach activities regarding this study and the health checkup project. In addition, we will also strive to provide opportunities for information meetings in municipalities outside the areas covered by this study. Third, we were unable to analyze the data according to sex or age. Although related to the second limitation, the small number of participants in the analysis made it difficult to analyze by age and sex. Although we adjusted for age and sex by including them as covariates, we believe that a more detailed analysis is needed because PhA and physical activity differ by age and sex [42,52,53]. Fourth, this study used the BIA method (Inbody S10), which is widely used as a simple and non-invasive method for measuring body composition. However, the validity of this body-composition method could vary depending on the characteristics of the target population (country studied, health status) and age [54,55]. In addition, in this study, SMI was calculated based on manuals and previous studies. However, studies in Europe have proposed methods of calculating SMI that could take sex and BMI into account [55]. In the future, if an appropriate method for calculating SMI in Japanese people is clarified, taking that method of calculation into account before analysis will also be necessary.

## 5. Conclusions

The results of this study clarify the association between PhA and MVPA in community-dwelling older adults. The results additionally suggest the need to examine interventions that increase MVPA and the effects thereof, in addition to the resistance training-centered intervention studies that have been previously conducted, in the effort to examine ways to increase PhA in older adults. Furthermore, future studies are needed to elucidate the causal relationship between PhA and MVPA through longitudinal research and interventional studies are needed to examine the effects of MVPA on PhA.

In addition, the results of this study suggest that the cutoff value that predicts high PhA in older adults is an MVPA of 19.7 min or more per day. Although further studies with a larger number of participants are needed to improve the accuracy of the analysis, the results may be used as information for recommending MVPA for older adults.

## Figures and Tables

**Figure 1 healthcare-12-00167-f001:**
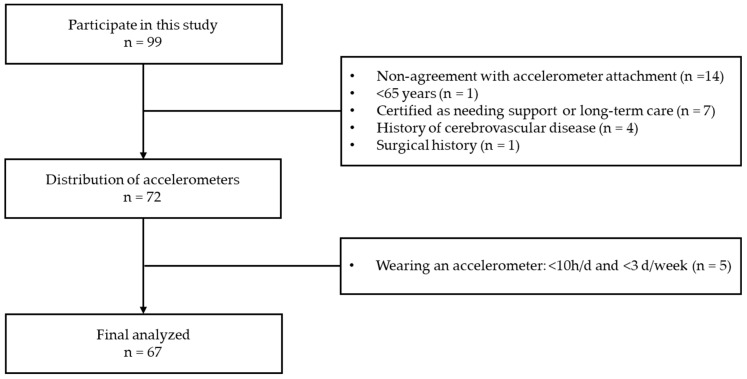
Flowchart of participant selection.

**Figure 2 healthcare-12-00167-f002:**
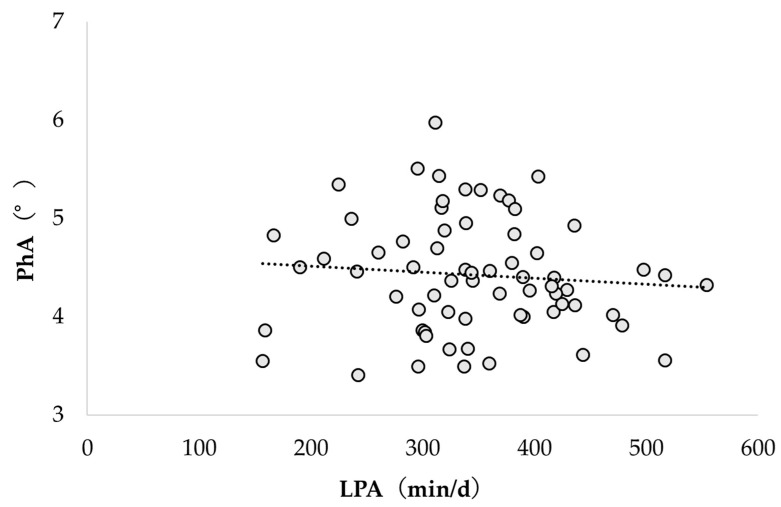
Correlation between PhA and LPA. PhA, phase angle; LPA, light-intensity physical activity.

**Figure 3 healthcare-12-00167-f003:**
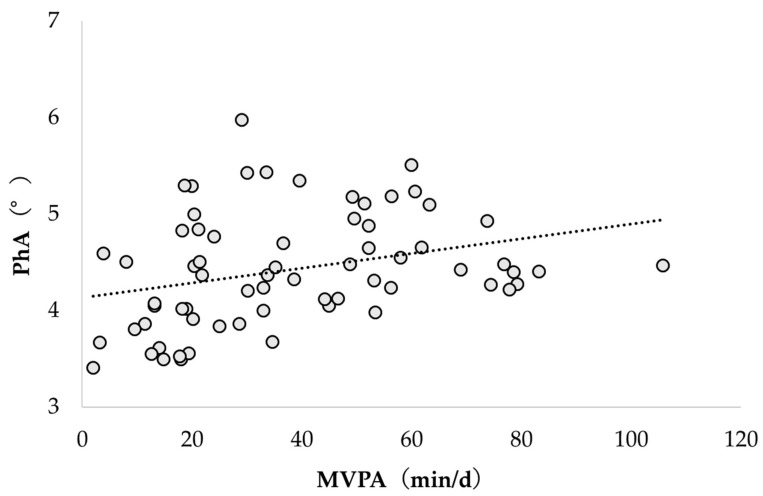
Correlation between PhA and MVPA. PhA, phase angle; MVPA, moderate to vigorous physical activity.

**Figure 4 healthcare-12-00167-f004:**
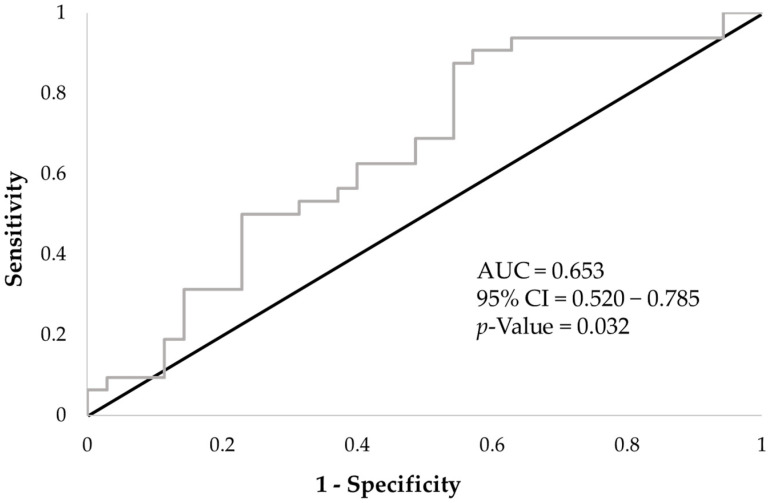
ROC curve of MVPA time for predicting high phase angle. AUC, area under the receiver operator characteristic curve; CI, confidence interval.

**Table 1 healthcare-12-00167-t001:** Participants’ characteristics.

Characteristic		Frequency (*n* = 67)
Age (years), mean ± SD		78.4 ± 5.5
Female (%)		83.6
Body mass index (kg/m^2^), mean ± SD		22.6 ± 3.4
Education (years), mean ± SD		13.1 ± 2.1
Living alone (%)		44.8
Accelerometer data	Wearing time (min/d), mean ± SD	850.9 ± 93.9
	LPA (min/d), mean ± SD	346.3 ± 84.0
	MVPA (min/d), mean ± SD	38.0 ± 23.2
Body Composition	Phase angle (°), mean ± SD	4.4 ± 0.6
	Low phase angle (%)	52.2
	SMI (kg/m^2^), mean ± SD	5.9 ± 0.8
Malnutrition (%)		11.9
Grip strength (kg), mean ± SD		22.1 ± 6.1
SPPB	Balance (score), median (first-third quartile)	4 (4-4)
	Gait speed (score), median (first-third quartile)	4 (4-4)
	Chair stand (score), median (first-third quartile)	4 (4-4)
	Total (score), median (first-third quartile)	12 (12-12)
Gait speed (m/s), mean ± SD		1.5 ± 0.4
Muscle quality, mean ± SD		44.0 ± 11.8

SD, standard deviation; LPA, light-intensity physical activity; MVPA, moderate-to-vigorous physical activity; SMI, skeletal muscle mass index; SPPB, short physical performance battery. Wearing time is the number of hours the accelerometer is worn per day. Malnutrition describes the percentage of participants judged to be malnourished by the GLIM criteria. Muscle quality is a result of Echo intensity.

**Table 2 healthcare-12-00167-t002:** Relationship between PhA and each variable.

Characteristic	Correlation Coefficient	*p*-Value
Age (years)	−0.343	<0.01
Sex	−0.526	<0.01
Body mass index (kg/m^2^)	0.204	0.099
LPA (min/d)	−0.089	0.475
MVPA (min/d)	0.305	0.012
SMI (kg/m^2^)	0.581	<0.01
Malnutrition	−0.176	0.155
Grip strength (kg)	0.645	<0.01
SPPB; balance tests (score)	−0.011	0.927
Gait speed (m/s)	−0.068	0.583
Muscle quality	−0.250	0.041

LPA, light-intensity physical activity; MVPA, moderate-to-vigorous physical activity; SMI, skeletal muscle mass index; SPPB, Short Physical Performance Battery.

**Table 3 healthcare-12-00167-t003:** Relationship between PhA and physical activity (Bayesian statistics).

	Model 1							Model 2					
	EAP	β	t	*p*-Value	95% Bayesian CI			EAP	β	t	*p*-Value	95% Bayesian CI	
LPA	−0.002	−0.228	−1.775	0.08	−0.003	0.001		−0.001	−0.113	−0.96	0.341	−0.002	0.001
MVPA	0.01	0.389	2.93	0.005	0.003	0.016		0.007	0.256	2.347	0.022	0.001	0.012
Age								−0.018	−0.173	−1.537	0.13	−0.041	0.006
Sex								0.097	0.064	0.28	0.781	−0.581	0.766
Body mass index								0.017	0.098	0.592	0.556	−0.039	0.073
SMI								0.12	0.175	0.712	0.479	−0.208	0.445
Malnutrition								−0.017	−0.014	−0.115	0.909	−0.318	0.265
Grip strength								0.034	0.358	1.64	0.107	−0.008	0.074
SPPB (balance tests)								−0.008	−0.004	−0.052	0.959	−0.331	0.309
Gait speed								−0.156	−0.102	−0.988	0.327	−0.464	0.161
Muscle quality								0.002	0.043	0.385	0.702	−0.008	0.012
	WAIC = 115.001							WAIC = 94.017					

EAP, expected a posteriori; CI, confidence interval; LPA, light-intensity physical activity; MVPA, moderate-to-vigorous physical activity; SMI, skeletal muscle mass index; SPPB, short physical performance battery. Muscle quality used Echo Intensity results. Model 1 is a model with no adjustment. Model 2 was adjusted for age, sex, BMI, SMI, grip strength, SPPB, gait speed, and muscle quality.

## Data Availability

The data supporting the findings of this study are available upon request from the corresponding author. The data are not publicly available because of privacy and ethical restrictions.

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
