# Peer review of "Relationship between Phase Angle and Physical Activity Intensity among Community-Dwelling Older Adults in Japan: A Cross-Sectional Study"

_healthcare, 2024, doi:10.3390/healthcare12020167_

Round 1

Reviewer 1 Report

Comments and Suggestions for Authors

A little more information regarding expected values and differences among healthy/clinical populations and race etc. would be helpful. Although a couple of meta analysis are included in refs, a few more citations would strengthen your argument and provide a better background.

more detail regarding recruitment, was this part of a regular checkup or outreach etc. 

line 109 should 'Metz' be 'Mets' metabolic equivalents.

in conclusion the addition of more refs could strengthen the argument to utilize PhA as an unobtrusive measure and would allow for a better understanding of the findings in the context of the current body of literature

limitations addressed but why not recruit until 100 subjects amenable to wearing device enroll?

Author Response

Dear Reviewer: 

We wish to re-submit the manuscript titled “Relationship between Phase Angle and Physical Activity Intensity among Community-Dwelling Older Adults in Japan: A Cross-Sectional Study.” The manuscript ID is healthcare-2770110.

We thank you and the reviewers for your thoughtful suggestions and insights. The manuscript has benefited from these insightful suggestions. I look forward to working with the reviewers to move this manuscript closer to publication in Healthcare.

The manuscript has been rechecked and the necessary changes have been made in accordance. Our revisions are denoted in red text in the manuscript. The responses to all comments have been prepared and please see the attachment. 

Thank you for your consideration. I look forward to hearing from you.

Sincerely,
Daiki Nakashima
Department of Rehabilitation, Faculty of Health Science, Naragakuen University, 3-15-1, Nakatomigaoka, Nara, Nara 631-8524, Japan
Tel.: +81-742-95-9800
Email: nakashima@naragakuen-u.jp

Reviewer 2 Report

Comments and Suggestions for Authors

Submission ID healthcare-2770110: This study explored the e the association between phase angle (PhA) and physical activity intensity in community-dwelling older Japanese adults. The research paper provides an interesting topic, but there are several issues to be addressed as follows.

Abstract

The abstract is not perfect. Please give more information on how to measure PA and PhA, and remove sentences about statistical analysis and focus on the results and conclusions.

Introduction

Line 48-50: the authors mention “A cross-sectional study of older Japanese adults (mean 48 age 75.2 years) reported that moderate-to-vigorous physical activity (MVPA) was associated with higher PhA [5].” I notice that the previous cross-sectional study has determined the relationship MVPA and PhA in Japanese. The results are the same as the current study. So, what is the novelty of the current study? This issue is very important.

Line 51: please uppercase the first letter.

Methods

(1) participants information including Figure 1 and sample size calculation should be placed in the first paragraph of the methods section.

(2) what’s the value of PhA is good? The lowest or the greatest value? Please state.

(3) please check the Line 135.

(4) the authors adjusted several factors when performing Bayesian statistics, such as BMI, grip strength, SPPB, and muscle quality. In addition to these factors, the reviewer think that diet has an important impact on PhA, it should be adjusted in this study.

(5) why the authors provided information on the association between the total physical activity and PhA?

(6) Whether it is possible to divide the participants into high and low PhA groups. Then, the authors explore the cut-off MVPA times for predicting high PhA.

Results

(1) figure 1 should be placed in the methods section.

(2) the description for tables and figures is too brief. Please provide more information about it.

Discussion

(1) the first paragraph is too short.

(2) other variables also should be discussed in this section such as age, sex, BMI, muscle quality, and so on.

Conclusions

Line 300-302: “This suggests the importance of examining the effects of MVPA in addition to the resistance training-centered interventions that have been examined in previous studies.” the importance for what? Please reword the sentence.

Author Response

(The authors gave the same response as above.)

Round 2

Reviewer 2 Report

Comments and Suggestions for Authors

Thank you for considering the previous comments and suggestions. The quality of this paper has been improved substantially. 

Several issue this time should be settled.

1. Is "Sample Size Calculator" or "Sample Size Calculation" right in line 92? 

2. The authors state in the paper "the target sample size was set to 110". However, this study only recruited 99 subjects, then 67 of them included the final analysis. Is that appropriate?

3. Line 165-167. Please note the source of the SMI formula, and its validity. 

Author Response

Dear Reviewer,

We wish to re-submit the manuscript titled “Relationship between Phase Angle and Physical Activity Intensity among Community-Dwelling Older Adults in Japan: A Cross-Sectional Study” The manuscript ID is healthcare-2770110.

We thank you and the reviewer for your thoughtful suggestions and insights. The manuscript has benefited from these insightful suggestions. 

The manuscript has been rechecked and the necessary changes have been made in accordance. The responses to all comments have been prepared and please see the attachment. 

Thank you for your consideration. I look forward to hearing from you.

Sincerely,

Daiki Nakashima

Department of Rehabilitation, Faculty of Health Science, Naragakuen University, 3-15-1, Nakatomigaoka, Nara, Nara 631-8524, Japan

Tel.: +81-742-95-9800

Email: nakashima@naragakuen-u.jp
